# Dataset of Public Objects in Uncontrolled Environment for Navigation Aiding

**Teng-Lai Wong** [1,*] , **Ka-Seng Chou** [1,2] , **Kei-Long Wong** [1] and **Su-Kit Tang** [1]

1 Faculty of Applied Sciences, Macao Polytechnic University, Macao SAR, China
2 Department of Computer Science and Engineering, Alma Mater Studiorum, University of Bologna, 47521 Bologna, Italy
* Correspondence: tenglaiwong@gmail.com

**Abstract:** Computer vision is a new approach to navigation aiding that assists visually impaired people to travel independently. A deep learning-based solution implemented on a portable device that uses a monocular camera to capture public objects could be a low-cost and handy navigation aid. By recognizing public objects in the street and estimating their distance from the user, visually impaired people are able to avoid obstacles in the outdoor environment and walk safely. In this paper, we created a dataset of public objects in an uncontrolled environment for navigation aiding. The dataset contains three classes of objects which commonly exist on pavements in the city. It was verified that the dataset was of high quality for object detection and distance estimation, and was ultimately utilized as a navigation aid solution.

**Keywords:** dataset; navigation aid; public object; deep learning; YOLOv4; computer vision





## 1. Summary

As stated in the 2020 ARVO Annual Meeting Abstract [1] released by the Association for Research in Vision and Ophthalmology (ARVO), the global population of people reported to be completely blind, moderately, and severely visually impaired was estimated to be 49.1 million, 221.4 million and 33.6 million, respectively. The increasing number of new visual impairment (VI) cases reported has drawn the attention of researchers in the community. The loss of eyesight is extremely inconvenient. The lives of visually impaired people are full of challenges in terms of accessibility and movability due to the low success rate of obstacle avoidance in public places.

In Macao, assisted-living facilities (ALFs), such as tactile paving and electronic audible traffic lights, are commonly installed in public areas by the government, enabling visually impaired people to recognize the routes of pavements and intersections. However, the coverage is not well planned in old districts. Therefore, personal navigation tools, such as white canes, guide dogs, etc., are commonly used. Recently, deep learning on computer vision has been promoted in many applications due to its high accuracy in object recognition, and it also has a high potential to be applied to near-distance in-front obstacle detection [2], allowing visually impaired people to travel independently and confidently.

In the literature, numerous computer vision-based studies have been conducted on the subject of assisting the visually impaired population. For example, an IoT device equipped with machine-learning techniques was designed to detect obstacles and alert visually impaired people audibly and tactilely [3]. Encouraging results were presented, and it is anticipated that a larger dataset will lead to improved performance. In [4], the authors proposed a machine learning-based obstacle detection device for the visually impaired

person, which concerned the different heights of an object (such as the chest, waist, and knee level of the user). In this study, the shape of hindrances was also considered, and promising results were presented. In addition, an Android smartphone application was presented to apply convolutional neural networks for obstacle detection [5]. The application was tested by an undisclosed dataset of images taken in the urban environment. In [6], a wearable obstacle detection device was proposed for use in indoor environments. Promising results were provided in that research, although the testing dataset was expected to have more object types. Furthermore, an IoT device mounted on a walking stick was proposed for visually impaired persons to avoid obstructions [7]. In this study, computer vision-based techniques were applied for object detection, and an ultrasonic sensor was used to measure object distance. The success of the aforementioned research relied heavily on a high-quality image dataset. However, publicly available image datasets suitable for obstacle detection or navigation aids in public and uncontrolled environments are still scarce in the research field. As a result, it is strongly urged that a dataset be created specifically for the purpose of aiding navigation for visually impaired individuals.

The main goal of this research is to develop a high-quality dataset of public objects in an uncontrolled environment for navigation aiding. Creating a high-quality dataset is crucial for the accuracy of prediction using deep learning technologies [8,9]. To create a dataset of public objects, a significant effort will be involved in determining their characteristics, as they are common, have different forms, and are distributed in crowded places. Thus, maintaining the quality of such a dataset becomes an issue. In this paper, the proposed dataset was targeted as a basis for object detection and distance estimation problems and will ultimately be used to aid navigation in crowded streets.

## 2. Data Description

This study released a dataset which contains public objects images and the annotation files of each image. The "images" folder contains 436 images (235 on a sunny day and 201 on a cloudy day) which were taken in two districts of Macao. The images were captured by a smartphone camera (12MP, 1/2.55" CMOS sensor, f/1.8 aperture, and 26mm focal length) at random angles, heights, and distances. A sample of the images is presented in Figure 1. Each image contains multiple target objects on different backgrounds (brick, concrete, and asphalt), and their relative sizes and orientations were different.

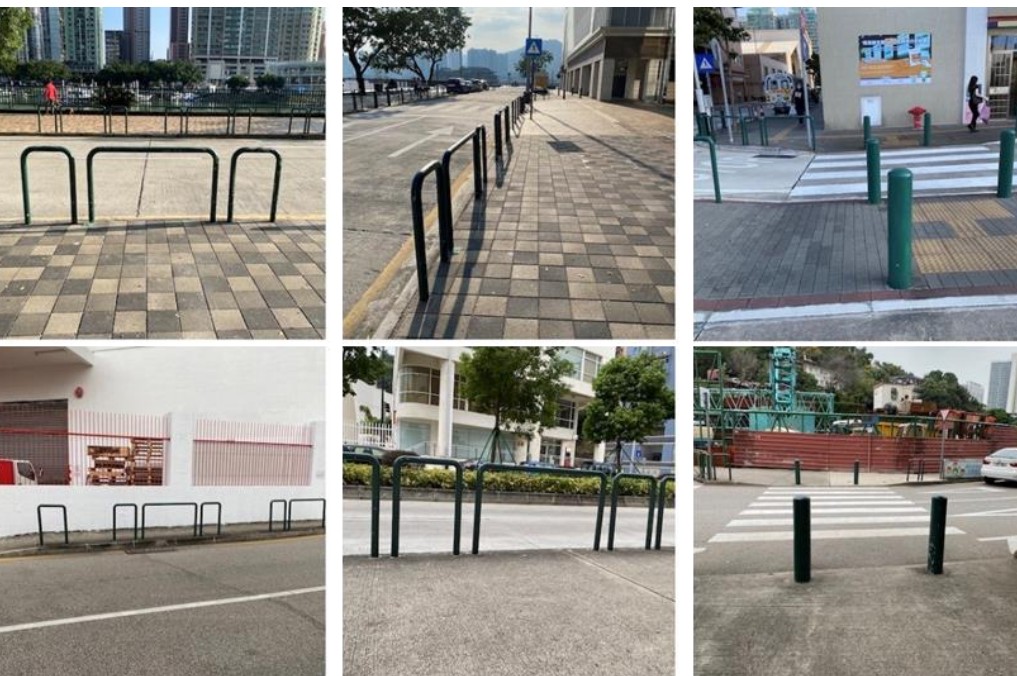

**Figure 1.** Images were taken at random angles, heights, and distances.

The folder named "labels" included PASCAL VOC annotations in XML files. Each annotation file contained the folder name, filename, path, image size, depth, class names, and the coordinates of bounding boxes of the target objects (Figure 2). Three classes of objects (label names) were created: *fence_short*, *fence_long*, and *pillar*. The number of labels varied from 1538 to 2938 depending on the labelling methods, which will be discussed in Section 3.4.

```xml
<annotation>
        <folder>fence-2021</folder>
        <filename>IMG_2014.jpeg</filename>
        <path>C:\Users\tlwong\Downloads\fence-2021\IMG_2014.jpeg</path>
        <source>
                <database>Unknown</database>
        </source>
        <size>
                <width>416</width>
                <height>416</height>
                <depth>3</depth>
        </size>
        0
        <object>
                <name>fence_long</name>
                <pose>Unspecified</pose>
                <truncated>0</truncated>
                <difficult>0</difficult>
                <bndbox>
                        <xmin>151</xmin>
                        <ymin>186</ymin>
                        <xmax>310</xmax>
                        <ymax>326</ymax>
                </bndbox>
        </object>
```

**Figure 2.** A screenshot of the annotation file.

## 3. Methods

### 3.1. Target Object Selection

In Macao, as the style of public objects on pavements, such as fences, pillars, fire hydrants, street name signposts, traffic signs, streetlights, etc., is regulated by the government, their dimensions, and colors are consistent (Figure 3). Among them, fences and pillars appear most frequently on pavements and are thus appropriate target objects.

Fences exist on the edge of the pavement in parallel with the road. Two types of fences are available in Macao: long fences, and short fences. Both are built by a U-shaped steel pole, painted green, and have the same height. Long fences are 142 × 86 cm (±1 cm), while short fences are 56 × 86 cm (±1 cm). They function as barriers that separate the pavement and the road and prevent pedestrians from walking out to the road intentionally or accidentally. Pillars are also located on the edges of pavements. They mainly appear in front of crossroads and zebra crossings to indicate where pedestrians may cross the road.

The number of fire hydrants and street name signposts on pavements is much lower than that of fences and pillars. Fire hydrants and street name signposts are sometimes hidden in the building or mounted on the wall. The styles of traffic signs vary depending on their meanings. The general spacing of streetlights is relatively large. In the area where the experiment was held, the spacing between streetlights was 30 m, but that of fences was less than 2 m. On the other hand, the complete exterior view of the streetlights can be captured only when they are at a distance.

Considering the spacing and exterior consistency, fences and pillars are the best candidates for selection as target objects of the experimental solution. Through these target objects, visually impaired people can elementarily localize themselves on the pavement, avoid walking out to roads accidentally, and know where they may cross roads.

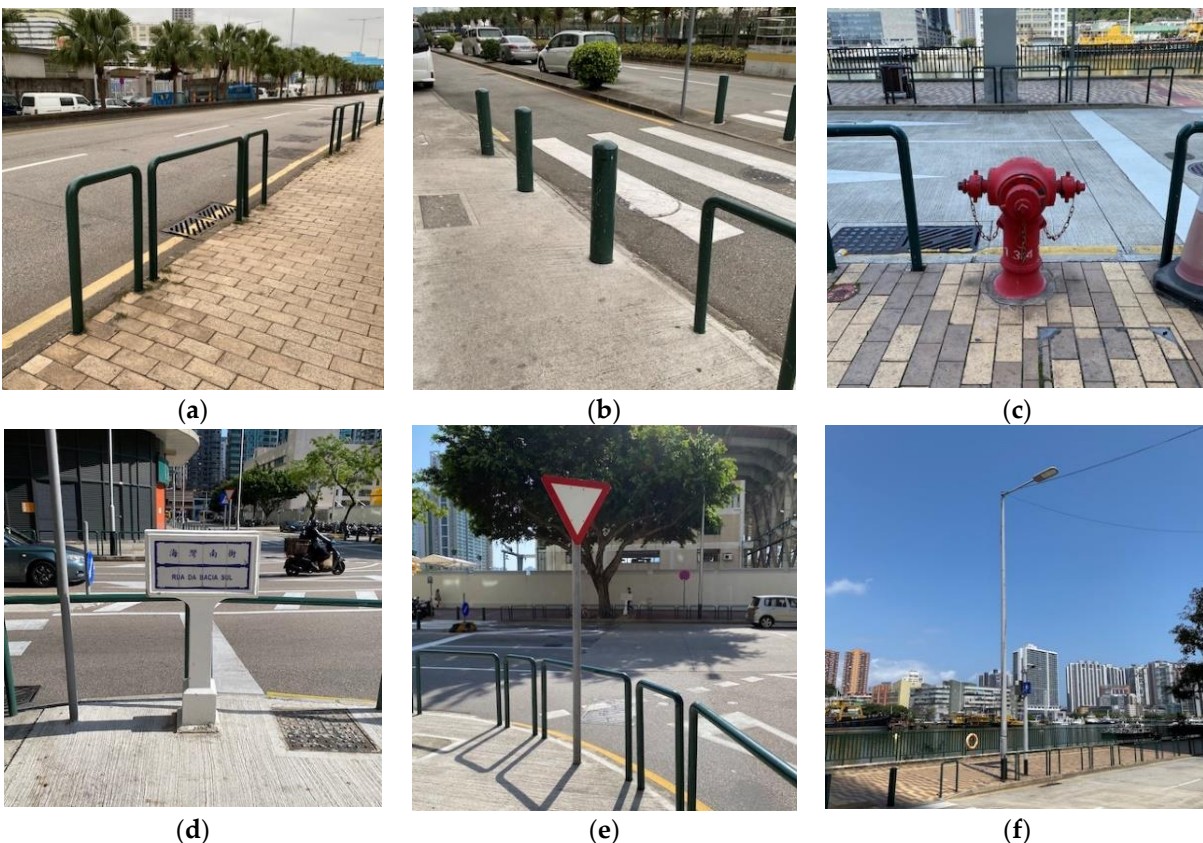

**Figure 3.** General public objects on the pavement in Macao: (**a**) fence; (**b**) pillar; (**c**) fire hydrant; (**d**) street signpost; (**e**) traffic sign; and (**f**) streetlight.

## 3.2. Model Selection

The current mainstream deep learning models for object detection can be divided into two categories: single-stage detector and two-stage detector [10]. The single-stage detector (YOLO, SSD, RetinaNet, etc.) calculates the image and generates detection results in one phase directly, without showing the process of the region proposal extraction. In contrast, the two-stage detector (R-CNN family [11–13]) first generates a sparse set of the bounding boxes from the image, and then makes corrections based on the region of the bounding box to improve the final detection results. Both detectors have their advantages in that the single-stage detector has a high inference speed and two-stage detector has high localization and recognition accuracy [14,15]. Therefore, a single-stage detector was selected in this research as it works efficiently on low-resolution images at low computing resources. It meets the requirements of a handy navigation aid solution, which was the final goal of this research.

Among the numerous candidates of single-stage detectors, YOLO (You Only Look Once) [16] is one of the most representative algorithms because of its high speed and accuracy. It was developed over a long period of time and several versions have been released [17,18]. YOLOv4 is one of the offshoots of YOLOv3 and achieves a faster and more accurate YOLO [19]. YOLOv4-tiny was proposed based on YOLOv4 to simplify the network structure and reduce the parameters. It is suitable for development on mobile and embedded devices. In this research, because a detector that required fewer computing resources and provided a faster inference speed was preferred, the YOLOv4-tiny was selected for the experiment.

### 3.3. Data Pre-Processing

In general, the images taken by cameras have an aspect ratio of 4:3 or 16:9. The native resolution of YOLOv4 is 416 × 416 pixels (aspect ratio = 1:1). Although Darknet accepts any image resolution (the only restriction is that the width and height can be evenly divided by 32), it will stretch the image to the exact size without preserving the aspect ratio prior to the training and inference processes. Resizing the non-squared image to a squared one causes shape distortion of the image and changes the visual characteristics and features of the target object [20]. The inference result will be affected if a model is trained on resized images with distorted objects. Therefore, the images for the dataset were taken in square mode in the experiment to keep the aspect ratio 1:1 from end-to-end to save time from the image-cropping process.

On the other hand, despite the fact that YOLOv4 is invariant to the size of the input image, there is a trade-off between speed, memory, and detection results. As the size of the image increases, the training and inference processes will take longer, and more GPU memory will be consumed. As the size of the image decreases, the training and inference processes will speed up, but it becomes difficult to detect small objects during the inference process. Training the YOLOv4-tiny model using original 4608 × 3456 JPG images on Darknet (complied to use GPU and OpenCV) for 10K iterations takes 11 h 44 min, whereas using 416 × 416 JPG images (quality = 75) reduces the time to only 1 h 53 min [21].

Considering that the target objects were large enough to be identified, the images in the experiments were down-sampled from 3024 × 3024 to 416 × 416, fitting the native resolution of YOLOv4, and converted to JPG, which is one of the YOLO-supported formats, in order to speed up the training and inference processes.

### 3.4. Labelling

Labelling was done manually using LabelImg [22]. LabelImg is a Python-based graphical image annotation tool. Qt is used for its graphical interface. Since LabelImg is open-source, cross-platform, and easy-to-use software, and YOLO only considers rectangular annotations, LabelImg was best suited for the experiment. The annotations can be exported in PASCAL VOC, YOLO, and CreateML formats. PASCAL VOC is an XML file which containing folder, filename, path, image size, depth, class names, and bounding box positions, whereas YOLO is a txt file which simply containing class IDs, bounding box center position, and its width and height. In this work, PASCAL VOC was selected because (1) PASCAL VOC is more popular than YOLO in the academic field; (2) PASCAL VOC can be used in different neural networks, and it is more general than YOLO; and (3) PASCAL VOC can be converted to YOLO. During the annotating process, the bounding boxes were drawn tight around the target objects and enclosed the entirety of the objects (Figure 4).

To further discuss how the labelling techniques affect the trained model, the experiment applied two dataset labelling methods: (1) the labelling of target objects within the effective distance, which was 2.4 m to 10 m in assumption, and ignored occluded objects as well as partially out-of-view objects; and (2) the labelling of all target objects in sight that included occluded, partially out-of-view, and far-distance objects, except objects which were too difficult to identify by the human eye. For fences partially out-of-view that could not be distinguished into long or short fences, *fence_short* and *fence_long* were labelled simultaneously, as shown in Figure 5. Table 1 shows the details of the labels in each class using the two labelling methods, and Figure 6 displays an example of the labelling in each dataset labelling method.

**Table 1.** Labels in each class by two labelling methods. The amounts of labels in method 2 were nearly double when far and partially out-of-view objects were labelled.

| Method | Image | *Fence_Short* | *Fence_Long* | *Pillar* | Total Amount |
|--------|-------|---------------|--------------|----------|--------------|
| 1 | 436 | 805 | 437 | 296 | 1538 |
| 2 | 436 | 1456 | 1077 | 405 | 2938 |

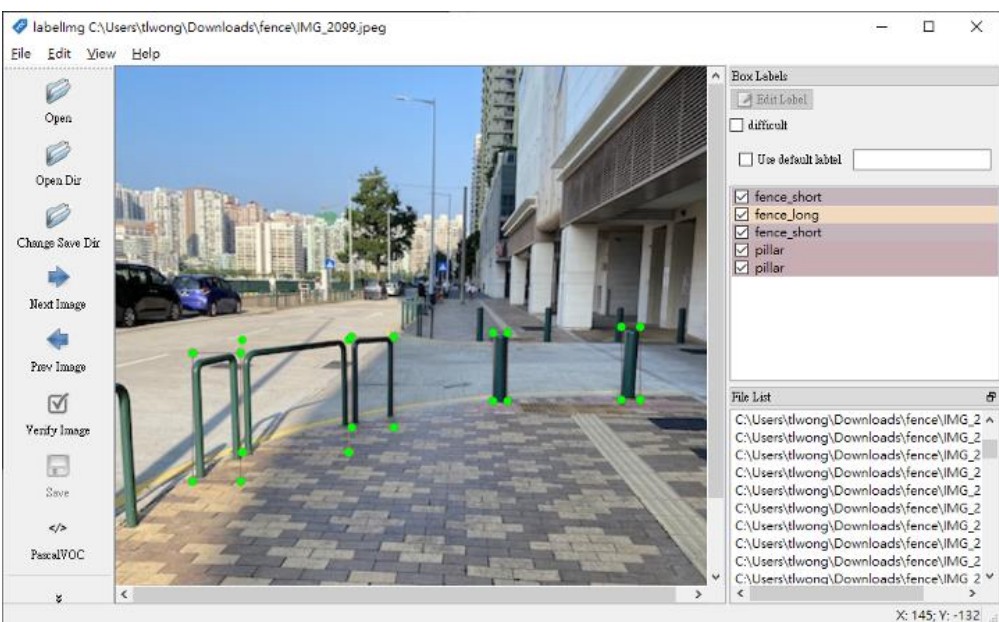

**Figure 4.** The annotations in LabelImg.

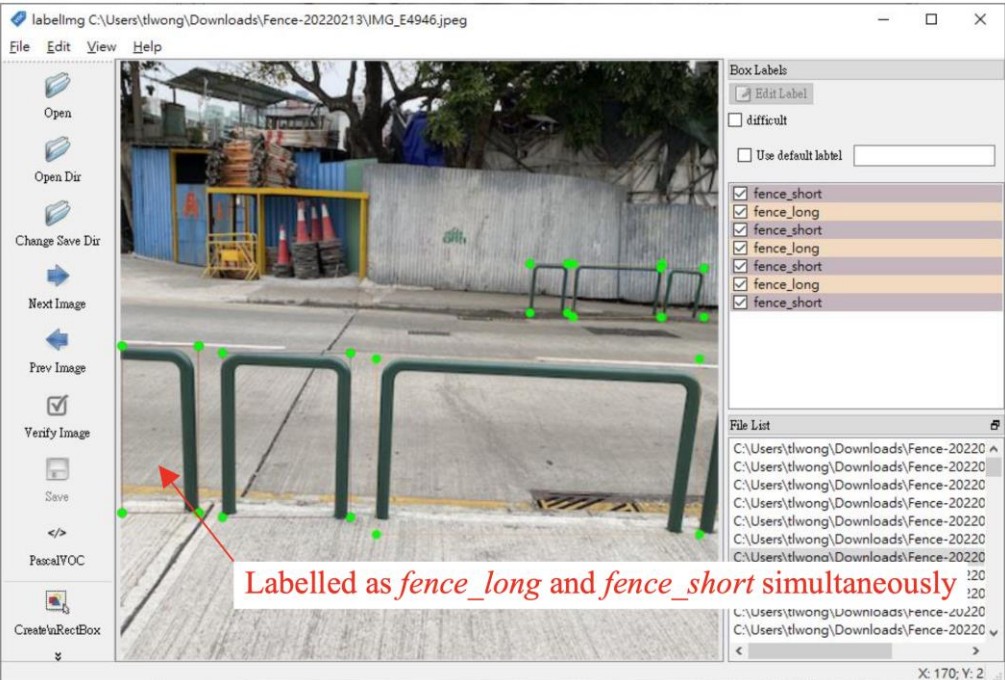

**Figure 5.** Example of labelling partially out-of-view objects. A partially out-of-view object was labelled as *fence_long* and *fence_short* simultaneously since it cannot be classified by a single image.

### 3.5. Data Validation

The goal of the experiment was to compare the YOLO models trained by different datasets in terms of their accuracy and inference outputs. YOLOv4-tiny, in three split ratios (training set: validation set = 70:30, 80:20, and 90:10), and the two dataset labelling methods were trained, respectively (Table 2), to compare the performance of the models using different datasets. The weights file was exported every 1000 iterations and the final weights file was exported at the 6000th iteration. The mAP@0.5 (mean average precision over IoU > 0.5) was calculated and logged every 100 iterations.

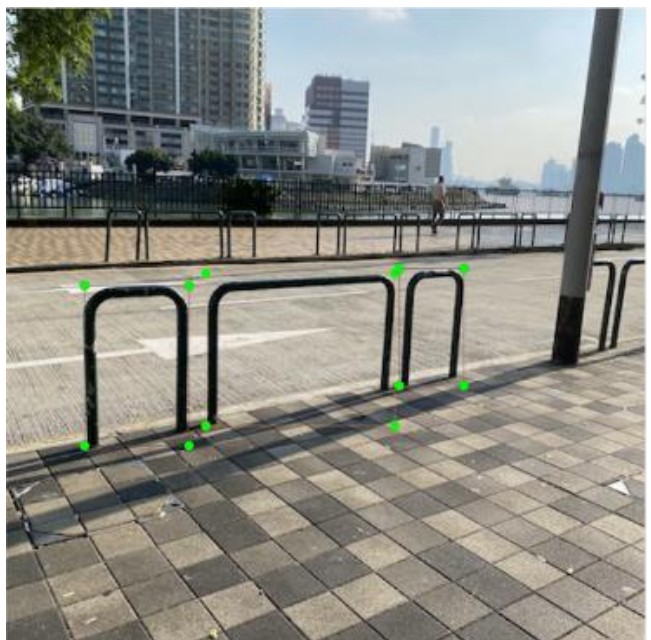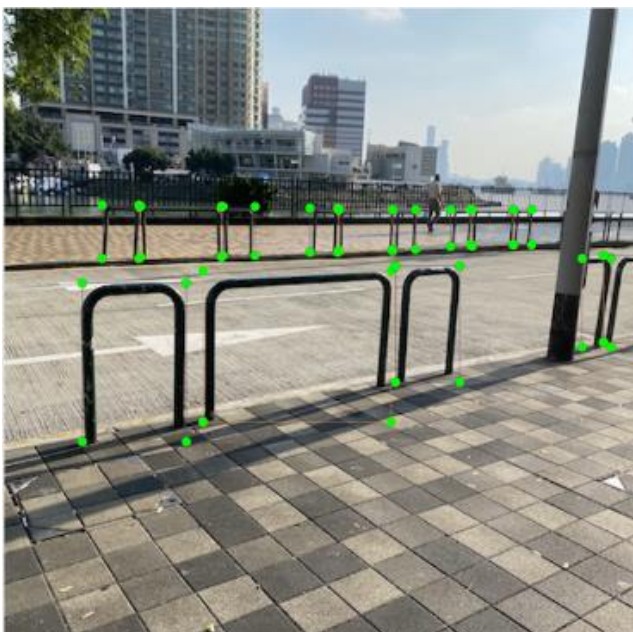

<div align="center">(<b>a</b>)</div>

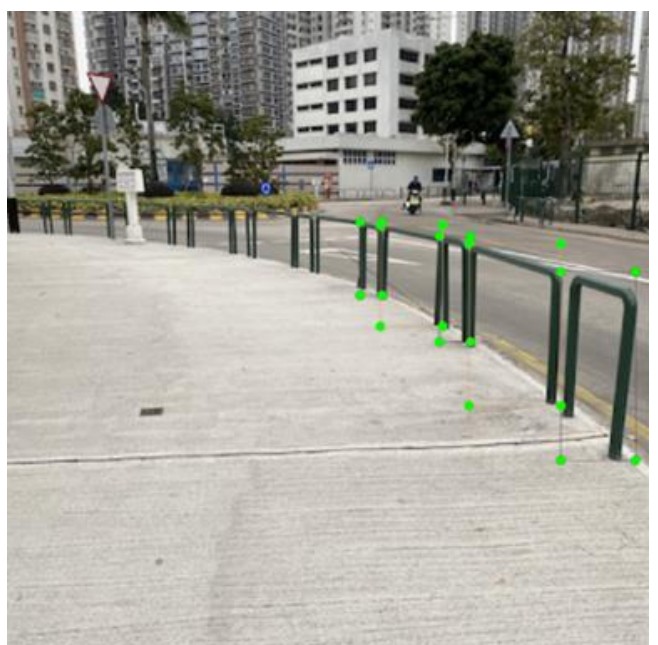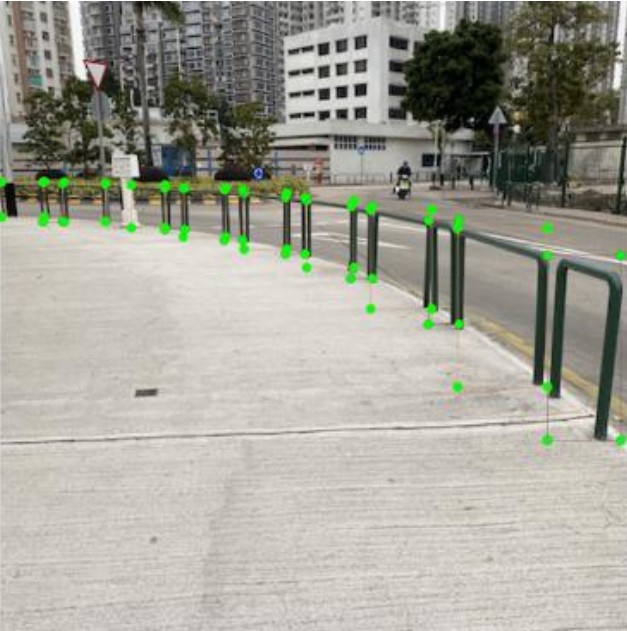

<div align="center">(<b>b</b>)</div>

**Figure 6.** The example of labelling in each labelling method: (**a**) labelling method 1; and (**b**) labelling method 2.

**Table 2.** Details of training models in the experiment.

| Training | Dataset (Images) | Model | Split Ratio | Dataset Labelling Method |
|---|---|---|---|---|
| #1 | 436 | YOLOv4-tiny | 70:30 | 1 |
| #2 | 436 | YOLOv4-tiny | 80:20 | 1 |
| #3 | 436 | YOLOv4-tiny | 90:10 | 1 |
| #4 | 436 | YOLOv4-tiny | 70:30 | 2 |
| #5 | 436 | YOLOv4-tiny | 80:20 | 2 |
| #6 | 436 | YOLOv4-tiny | 90:10 | 2 |

The average loss and mAP curves over 6000 iterations for each model are shown in Figure 7 and the comprehensive comparison of the training results of YOLO models using different datasets is shown in Table 3. All models provided a satisfying performance (mAP > 80%, avg. IoU > 70% and F1-score > 0.85) and could be directly used for the detection of objects. The results were attributed to the simple and standard exteriors of the target objects and the fact that only three classes of objects were available in the experiment. The more classes there are to be detected, the more iterations and the greater an amount of time the training process takes.

**Table 3.** Comparison of the training results.

| Training | Model | Class 0 [1] AP (%) | Class 1 [1] AP (%) | Class 2 [1] AP (%) | mAP@ 0.5 (%) | Avg. Loss (%) |
|---|---|---|---|---|---|---|
| #1 | YOLOv4-tiny | 93.44 | 92.62 | 65.09 | 83.72 | 0.061 |
| #2 | YOLOv4-tiny | 97.47 | 95.94 | 84.62 | 92.68 | 0.061 |
| #3 | YOLOv4-tiny | 95.00 | 95.01 | 87.79 | 92.60 | 0.073 |
| #4 | YOLOv4-tiny | 90.75 | 92.91 | 57.95 | 80.54 | 0.124 |
| #5 | YOLOv4-tiny | 93.02 | 90.76 | 74.83 | 86.20 | 0.133 |
| #6 | YOLOv4-tiny | 91.08 | 94.26 | 72.39 | 85.91 | 0.149 |

| Training | Model | Precision | Recall | F1-Score | Avg. IoU (%) | |
|---|---|---|---|---|---|---|
| #1 | YOLOv4-tiny | 0.79 | 0.90 | 0.84 | 72.13 | |
| #2 | YOLOv4-tiny | 0.80 | 0.95 | 0.87 | 72.75 | |
| #3 | YOLOv4-tiny | 0.80 | 0.92 | 0.85 | 71.94 | |
| #4 | YOLOv4-tiny | 0.82 | 0.86 | 0.84 | 73.88 | |
| #5 | YOLOv4-tiny | 0.83 | 0.89 | 0.86 | 74.42 | |
| #6 | YOLOv4-tiny | 0.90 | 0.87 | 0.88 | 79.31 | |

[1] Class 0 = *fence_short*, Class 1 = *fence_long*, Class 2 = *pillar*.

### 3.5.1. Comparison between Dataset Labelling Methods

Comparing the training results of the dataset of labelling method 1 (#1, #2, and #3) and method 2 (#4, #5, and #6), shown in Table 3, the model of method 1 had a better mAP. Although the dataset labelled by method 2 was nearly twice the size of method 1, the added labels were mostly at a distance, and were barely distinguishable background objects. It was logical, therefore, that the mAP of method 2 decreased. The average losses after 6000 iterations increased because the model required more iterations to finish the training process when a larger dataset was used for the training.

However, examining the inference outputs of the YOLOv4-tiny models (#2, #3, #5, and #6) which were trained by the datasets in two labelling methods for the same iterations (Figure 8), the detection of method 2 provided more details and showed more comprehensive information of the environment, and although some details might be unnecessary to some users, these could be filtered out in the latter process because only near-front objects were considered for the proposed solution. Partial objects out-of-view could also be detected. The classification of long and short fences of partial objects might be incorrect, but this did not affect the application as a navigation aid.

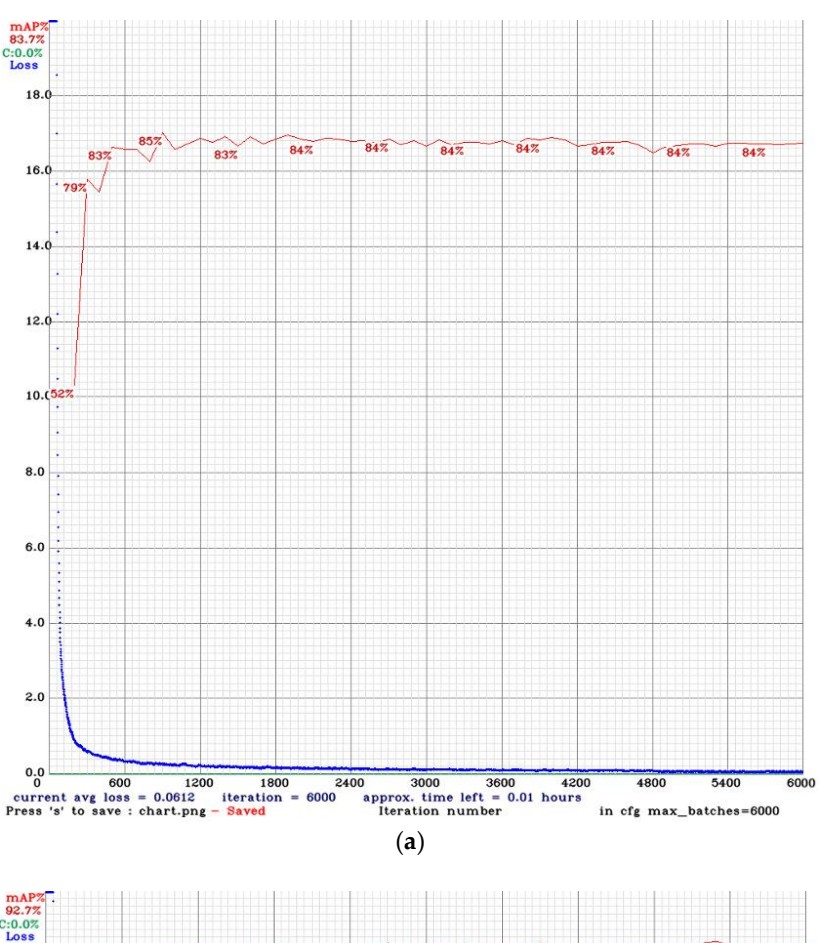

(**a**)

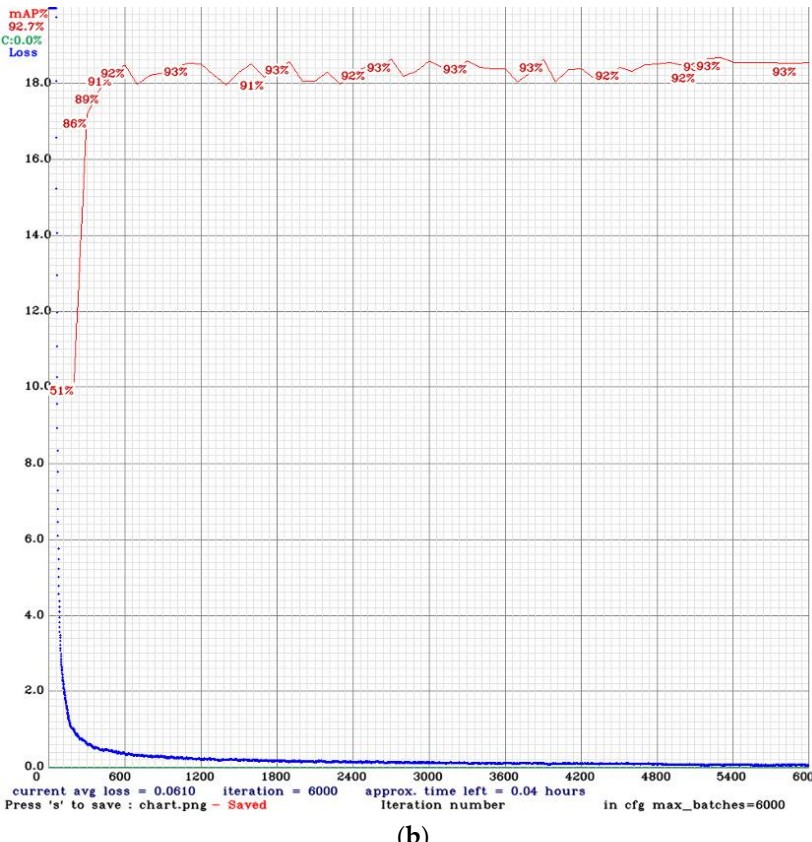

(**b**)

**Figure 7.** *Cont.*

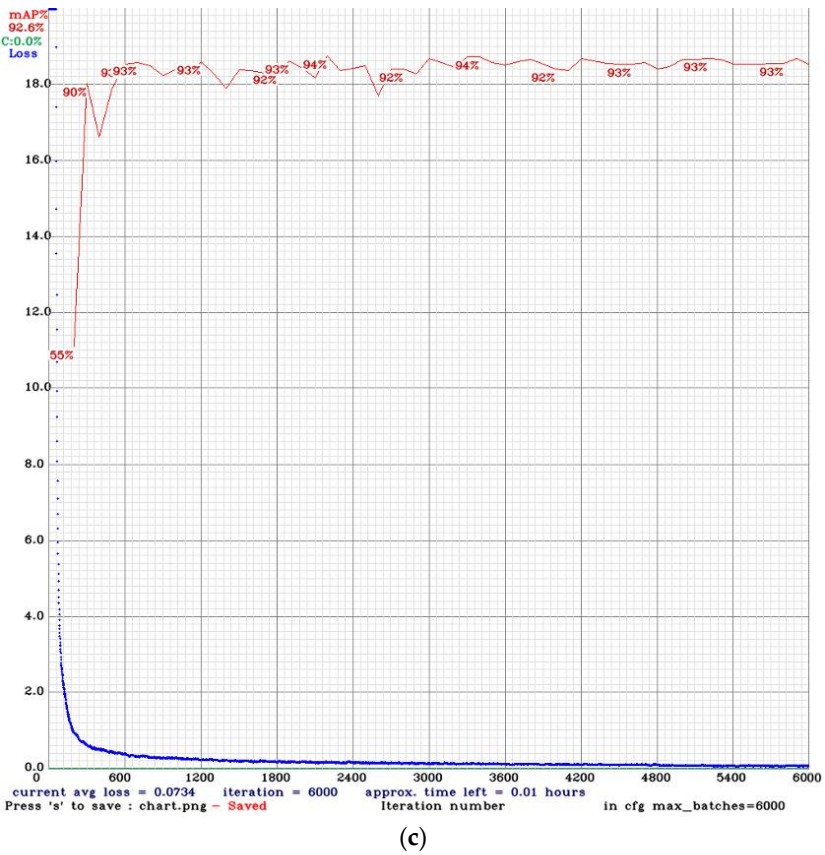

(**c**)

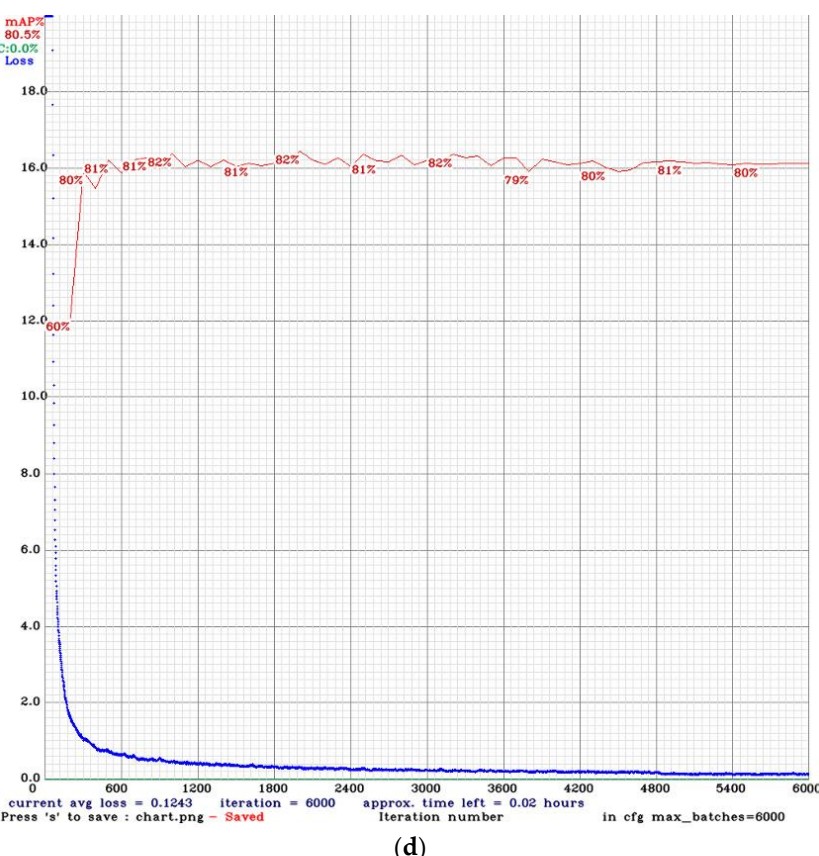

(**d**)

**Figure 7.** *Cont.*

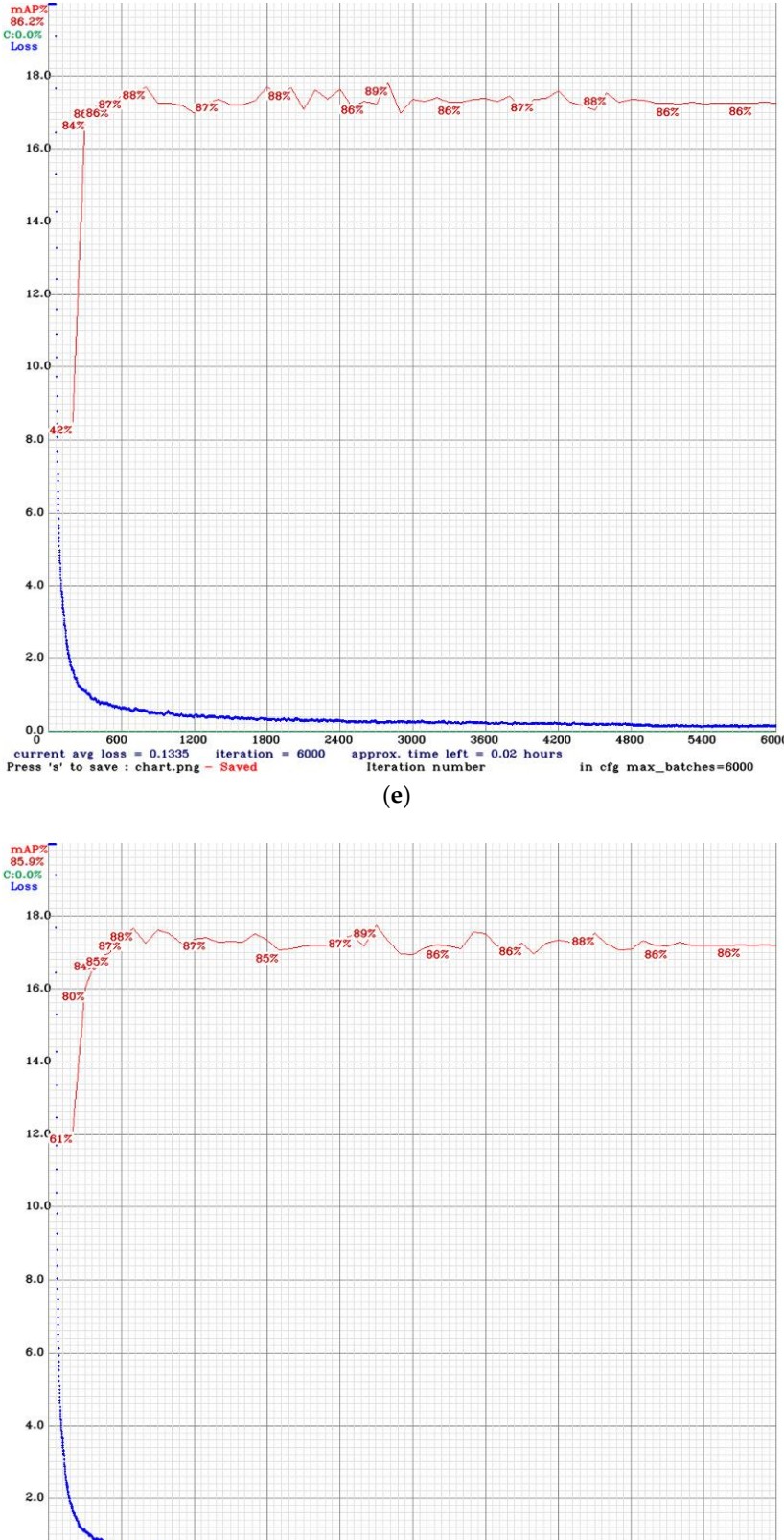

**Figure 7.** Average loss and mAP chart of each training. (**a**) Training #1. (**b**) Training #2. (**c**) Training #3. (**d**) Training #4. (**e**) Training #5. (**f**) Training #6.

### 3.5.2. Comparison among Different Dataset Split Ratios

YOLOv4-tiny models using different dataset split ratios (70:30, 80:20, and 90:10) are compared in Table 3. Since there is no absolute answer to the best dataset split ratio, as the ratio depends on empirical truth, through the experiment we looked for an appropriate dataset split ratio. The results showed that the models trained by 80:20 (#2, #5) and 90:10 (#3, #6) had better results, and 80:20 achieved the maximum mAP in the two groups (dataset labelling method 1 and method 2). The 70:30 (#1, #4) ratio had a lower mAP as the training set was smaller, so the model had insufficient data to train. Thus, 70:30 was not recommended to be applied due to the size of the dataset.

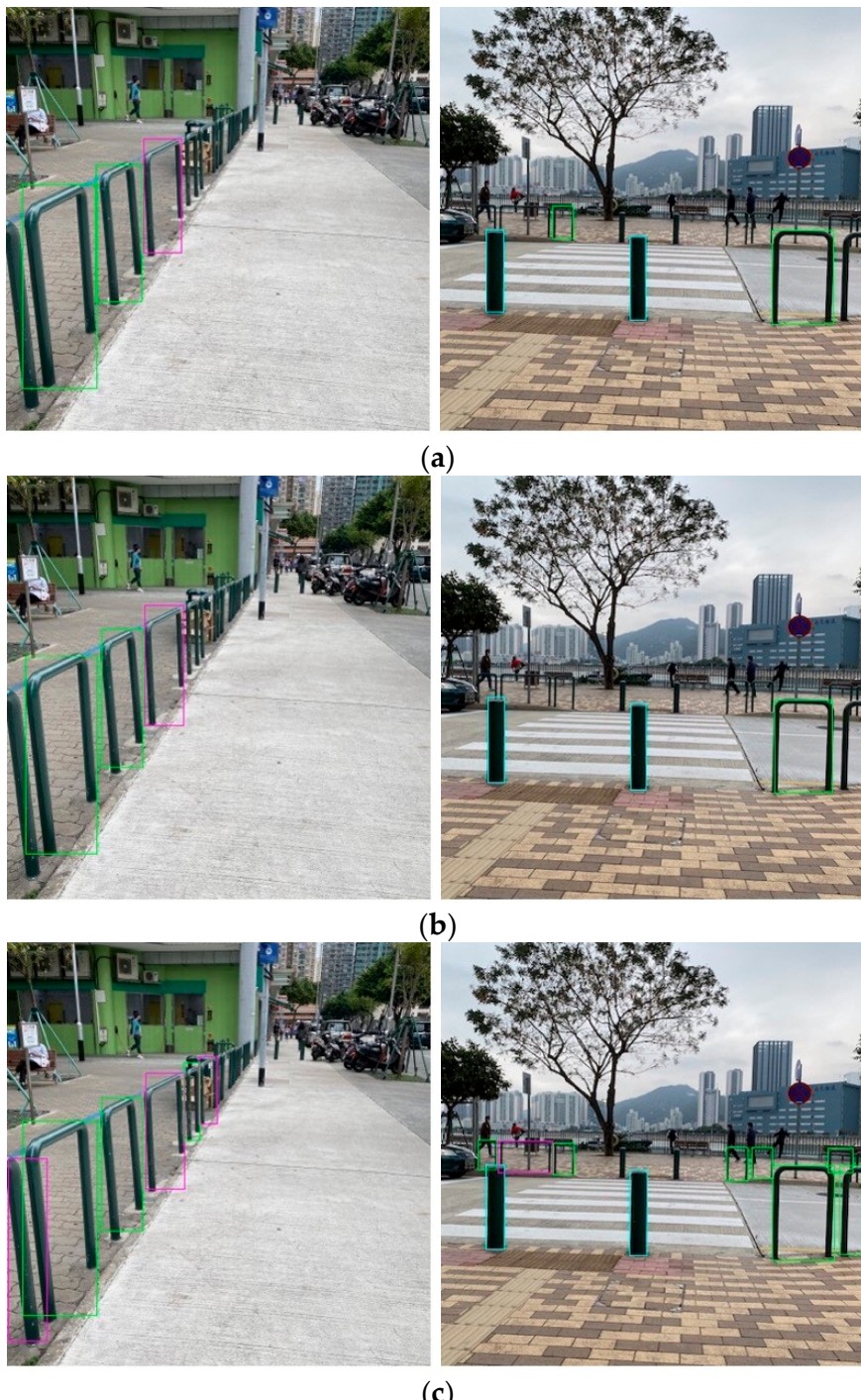

(**a**)

(**b**)

(**c**)

**Figure 8.** *Cont.*

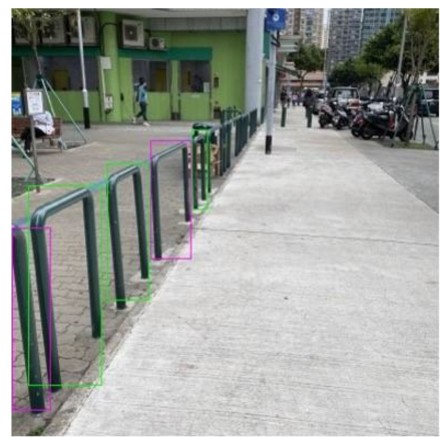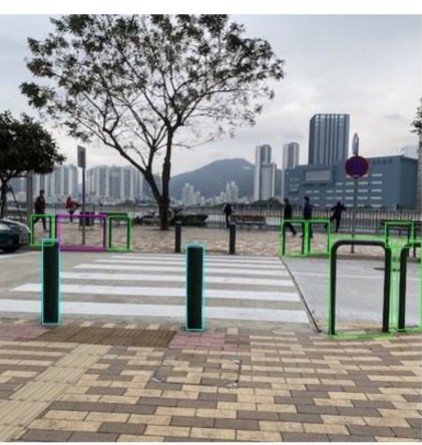

(**d**)

**Figure 8.** Example of detections of YOLOv4-tiny between difference labelling methods. The detection of method 2 provided more details and showed more comprehensive information of the environment around. (Class 0 = *fence_short* (green), Class 1 = *fence_long* (magenta), Class 2 = *pillar* (cyan)): (**a**) YOLOv4-tiny, labelling method 1, split ratio = 80:20 (training #2); (**b**) YOLOv4-tiny, labelling method 1, split ratio = 90:10 (training #3); (**c**) YOLOv4-tiny, labelling method 2, split ratio = 80:20 (training #5); and (**d**) YOLOv4-tiny, labelling method 2, split ratio = 90:10 (training #6).

The effect of an unbalanced dataset was revealed in the training results. Comparing the results in Table 3, the AP (average precision) of class 0 (*fence_short*) and class 1 (*fence_long*) was significantly higher than class 2 (*pillar*) due to the labels in *fence_short* and *fence_long* were 2 to 3 times much than labels in *pillar* (Table 1).

### 3.5.3. Distance Estimation Results

To verify that the proposed dataset can be used in navigation-aiding applications, the best-trained model (YOLOv4-tiny, dataset split in 80:20, labelling method 2) was adopted for two distance-estimation models:

(1) Position-based estimation: when an object is on the same plane as the camera, the position of the object in pixel coordinates can convert to real-world coordinates [23]. The distance between the object and the camera can be estimated by a single image from a monocular camera, of which the Field of View (FOV) and the height are known;

(2) Size-based estimation: since the class of the object in the image can be identified, and the size of the public object in each class is standard, the distance can be estimated using the size information of the detected object [24]. The experiment preferred to use the height of the target object instead of the width since the projection width of the object in the image varies along with the oblique angle to the camera axis. Meanwhile, the height is nearly unchanged assuming that the lens distortion is negligible.

Nine images taken at known poses were adopted in the object-detection inference and distance-estimation processes (Figure 9). The heights of the camera and the distances to the insight target objects were measured while capturing the images. Forty public objects were detected and located in the experiment. The comparison of ground truth and estimated distance in two distance-estimation models is illustrated in Figure 10.

The overall performance of the position-based model at short distances was better, as the mean absolute percentage error (MAPE) within the effective distance (assumed to be 10 m) of the position-based model was 6.18%, but the MAPE of the size-based model was 6.59%.

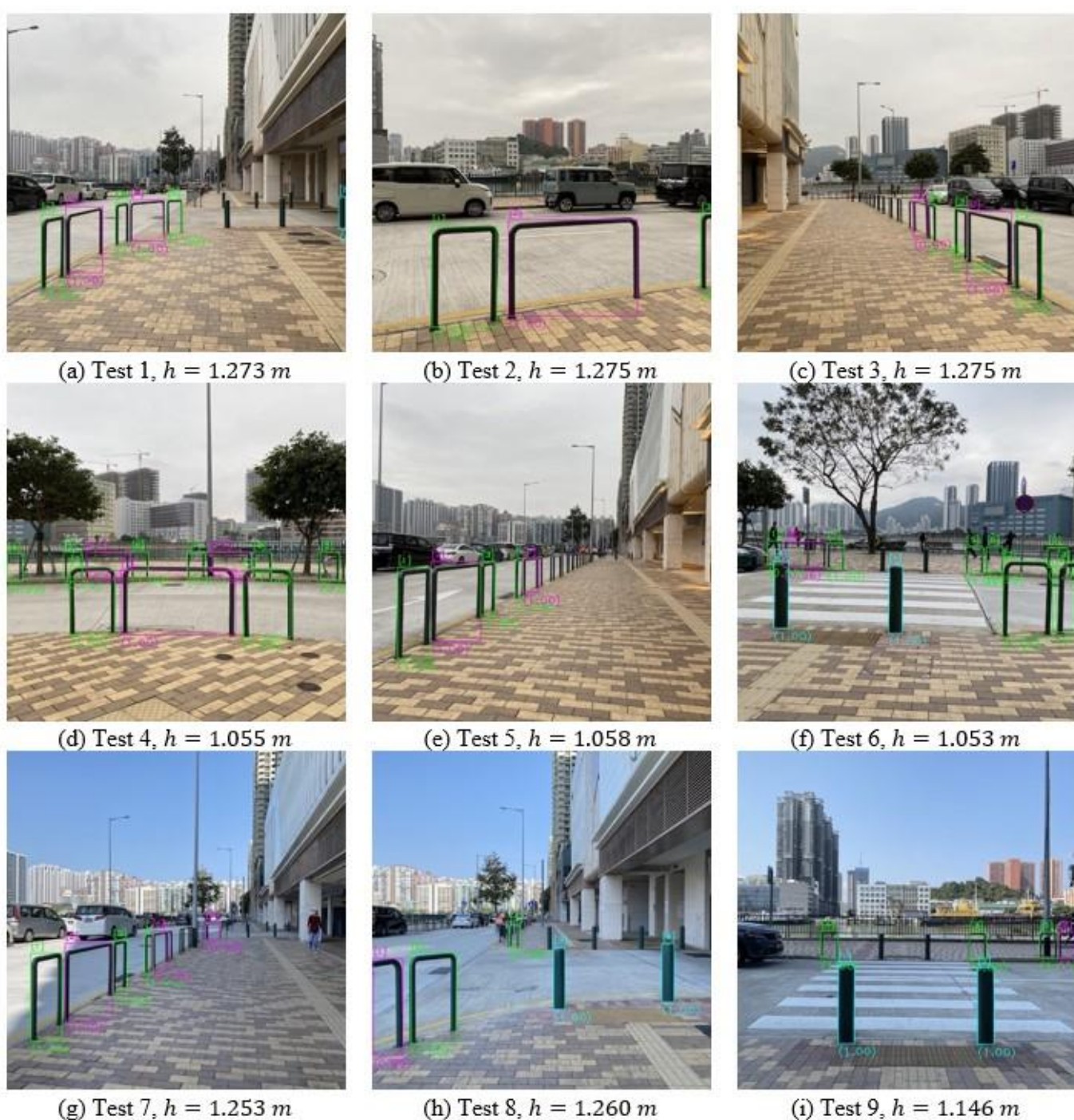

(a) Test 1, $h = 1.273\ m$      (b) Test 2, $h = 1.275\ m$      (c) Test 3, $h = 1.275\ m$

(d) Test 4, $h = 1.055\ m$      (e) Test 5, $h = 1.058\ m$      (f) Test 6, $h = 1.053\ m$

(g) Test 7, $h = 1.253\ m$      (h) Test 8, $h = 1.260\ m$      (i) Test 9, $h = 1.146\ m$

**Figure 9.** Detection output in the experiment (YOLOv4-tiny).

However, when taking the objects out of the effective distance into account, the size-based model provided a much better result. The MAPE of the size-based model was only 5.96% whereas, the MAPE of the size-based model climbed to 14.03%. The size-based model showed the capability of estimating distances in a dynamic range. One of the limitations of the size-based model is that the detected object must be completely detected. The misunderstanding of the sizes of occluded and partially out-of-view objects by the deep learning model will lead to a relatively large error.

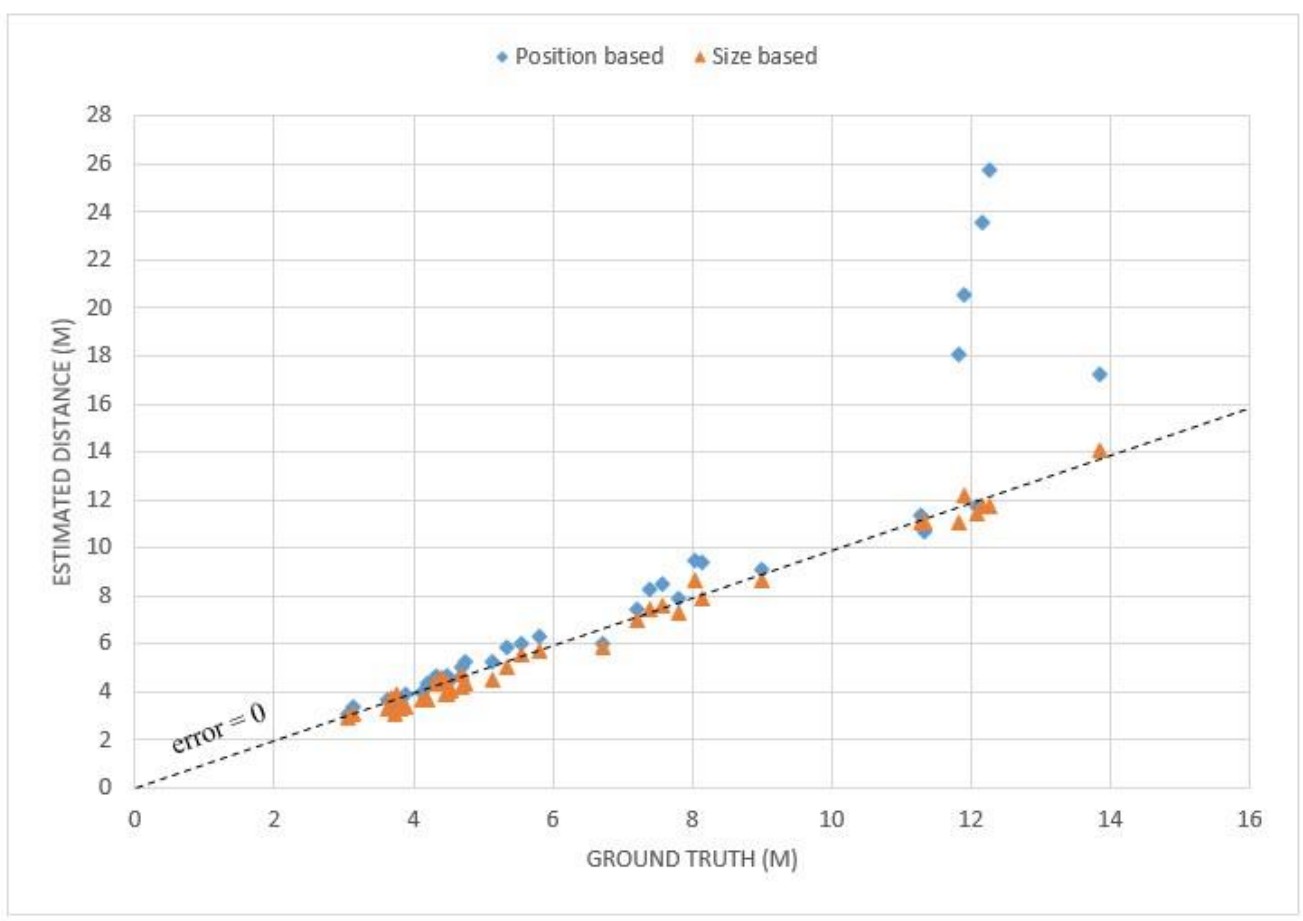

**Figure 10.** Position-based vs. size-based distance estimation model.

## 4. Conclusions

This research presented a specific dataset on public objects for navigation aids. The dataset consisted of 436 images which contained three classes of objects (long fence, short fence, and pillar) and 2938 labels in total. The selected public objects commonly exist on pavements in the city and can be utilized as reference objects for visually impaired people. To elaborate on the quality of the dataset, several performance tests were conducted for the recognition training and inference, choosing the appropriate labelling method and dataset splitting ratio. The dataset provided a YOLOv4-tiny model which had a mAP = 86.20%, F1 score = 0.86, and an average IoU = 74.42%.

The addition of more high-quality data in the dataset will increase the accuracy of the predictions made by using the deep learning method. One suggested improvement is to enlarge the dataset to more than 2000 labels for each class. To enhance the model's robustness, other common public objects, such as fire hydrants, street name signposts, traffic signs, streetlights, and especially humans and cars, should be included in the object detection dataset.

**Author Contributions:** Conceptualization, T.-L.W. and S.-K.T.; methodology, T.-L.W. and K.-S.C.; software, T.-L.W.; validation, T.-L.W., K.-S.C. and K.-L.W.; formal analysis, T.-L.W.; investigation, T.-L.W.; resources, T.-L.W.; data curation, T.-L.W.; writing—original draft preparation, T.-L.W.; writing—review and editing, K.-S.C. and K.-L.W.; visualization, T.-L.W. and K.-S.C.; supervision, S.-K.T.; project administration, S.-K.T.; funding acquisition, S.-K.T. All authors have read and agreed to the published version of the manuscript.

**Funding:** This research received no external funding.

**Institutional Review Board Statement:** Not applicable.

**Informed Consent Statement:** Not applicable.

**Data Availability Statement:** The dataset presented in this paper is openly available in Zenodo at https://doi.org/10.5281/zenodo.7542910 (accessed on 19 January 2023).

**Acknowledgments:** This work is supported in part by the research grant (No.: RP/ESCA-04/2020) offered by Macao Polytechnic University.

**Conflicts of Interest:** The authors declare no conflict of interest.

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
