# Peer review of "Dataset of Public Objects in Uncontrolled Environment for Navigation Aiding"

_data, 2022_

Round 1

Reviewer 1 Report

Some moderate changes in the English language should be made. In some cases lack of usage of articles - in the abstract and in the text e.g. "in an uncontrolled environment", "of a high quality" sound better with articles. Check the text and pay attention to articles and other mistakes. 

Some of the mistakes:

The sentence line 16-17 "The loss (....) and terrifying disabled" - reformulation needed. Line 34 "will" - double "l", line 35 "utilised to" - used for. 

LinÄ™ 48 and 49 - Pillar in line 48, in line 49 "They" related to the subject of the previous sentence "pillar" so the pronoun should be"it". 

LinÄ™ 52 fire hydrantS would be better because the rest of the obstacles mentioned are in plural form. 

The number of references is very low. 

Line 59-62 - reformulate the sentence;

Line 81 Resizing A non-squared image to a squared ONE.

Check the whole text and correct the mistakes.

There are few references. It is a scientific article so more references are needed because mostly the research undertaken by the authors is depicted. The state of research in this topic is missing - it can be shortly mentioned. The references to other authors need to be incorporated into the text and the bibliography needs to be completed. 

Author Response

  1. Some sentences have been revised while fixing the grammar mistakes.
  2. Add a paragraph in introduction to mention the state and related works of the research topic (line 28).
  3. Add more references in the bibliography (line 236-278).

Reviewer 2 Report

It is recommended to increase the number of references by including more scientific publications.

It is preferable to justify the selection of the labeling program Label Img and Machine Learning Model YOLOv4.

Author Response

  1. Additional references have been added (line 236-278).
  2. A subsection is added to justify the selection of YOLOv4 (line 94) and LabelImg (line 139).

Reviewer 3 Report

The paper discusses the creation of a dataset related to objects placed in streets and the execution of some experiments with the use of YOLO.

From my perspective, the paper is very simple and does not expose the necessary theoretical background as well as the apporpriate description to excuse a Journal publication. I cannot see any significant problem that the paper tries to solve and it seems that it is just an application effort. The research orientation of the paper is limited, thus, i cannot suggect its acceptance.

Author Response

The dataset is designed for distance estimation for navigation in streets by visual impair people. As public objects are commonly seen and have a standard size in the streets, they are good to be used in obstacle detection and distance estimation in a city scale outdoor environment. The utlimate goal the complete research is to develop a handly and low price navigation aid which can be used in an uncontrolled enviroment. The success of the aforementioned research relies heavily on a high-quality image dataset. However, publicly available image datasets suitable for obstacle detection or navigation aids in public and uncontrolled environments are still scarce in the research field. As a result, it is strongly urged that a dataset be created specifically for the purpose of aiding navigation for visually impaired individuals. 

Reviewer 4 Report

Excellent work on the manuscript.

I have used labellmg before many times, excellent tool.

Author Response

Some sentences have been revised while fixing the grammar mistakes

Round 2

Reviewer 3 Report

I appreciate the efforts of the authors to improve the quality of the paper, however, i still cannot discern any significant theoretical contribution and its evaluation. The paper presents the application of already defined deep learning models. So, i 'll stay at my decision at the previous review round and pass the floor to the reamining reviewers. 

Author Response

Thank you very much for your comments.

This paper seems doesn't have much innovation because it focuses on the work of creating a dataset. In fact, most of the datasets existing for distance estimation tasks are general objects on the road like humans, cars, and safety cones, they were created for self-driving. This paper presents the idea of creating a dataset specifically for visually impaired people and public objects in an uncontrolled environment in a particular city. This type of dataset is insufficient and of course our method can be generalized to any other international cities. Creating this kind of dataset is a predecessor task of our proposed model. We are going to utilize the dataset to process distance estimation on a monocular camera worn by visually impaired people.